# In Vitro and In Vivo Sucrosomial^®^ Berberine Activity on Insulin Resistance

**DOI:** 10.3390/nu14173595

**Published:** 2022-08-31

**Authors:** Maria Giovanna Lupo, Elisa Brilli, Virginia De Vito, Germano Tarantino, Stefania Sut, Irene Ferrarese, Giovanni Panighel, Daniela Gabbia, Sara De Martin, Stefano Dall’Acqua, Nicola Ferri

**Affiliations:** 1Department of Medicine, University of Padova, 35128 Padova, Italy; 2R&D Department, PharmaNutra S.p.A., 56122 Pisa, Italy; 3R&D Department, Alesco S.r.l., 56122 Pisa, Italy; 4Department of Pharmaceutical and Pharmacological Sciences, University of Padova, 35131 Padova, Italy

**Keywords:** Sucrosomial^®^, insulin, OGTT, gastrointestinal absorption, berberine, AMPK

## Abstract

*Background*: Berberine is a natural alkaloid with hypoglycemic properties. However, its therapeutic use is limited by a very low oral bioavailability. Here we developed a new oral formulation of berberine based on Sucrosomial^®^ technology and tested its effect on insulin resistance. *Methods*: Sucrosomial^®^ berberine was first tested in vitro in the hepatoma cell line Huh7 to assess its effect on proteins involved in glucose homeostasis and insulin resistance. The pharmacokinetics and efficacy on insulin resistance were then studied in C57BL/6 mice fed with standard (SD) and high-fat diet (HFD) for 16 weeks and treated daily during the last 8 weeks with oral gavage of Sucrosomial^®^ berberine or berberine. *Results*: Sucrosomial^®^ berberine did not affect Huh7 cell viability at concentrations up to 40 µM. Incubation of Huh7 with 20 µM of Sucrosomial^®^ and control berberine induced glucokinase (GK) and the phosphorylation of 5′-adenosine monophosphate (AMP)-activated protein kinase (AMPK), both known targets for the control of insulin resistance. In vivo, we observed an 8-fold higher plasma concentration after 3 weeks of oral administration of 50 mg/kg/day of Sucrosomial^®^ formulation compared to berberine. HFD, compared to SD, induced insulin resistance in mice as determined by oral glucose tolerance test (OGTT). The treatment with a 6.25 mg/kg/daily dose of Sucrosomial^®^ berberine significantly reduced the area under the curve (AUC) of OGTT (73,103 ± 8645 vs. 58,830 ± 5597 mg/dL × min), while control berberine produced the same effects at 50 mg/Kg/day (51518 ± 1984 mg/dL × min). Under these conditions, the two formulations resulted in similar berberine plasma concentration in mice. Nevertheless, a different tissue distribution of metabolites was observed with a significant accumulation of reduced, demethylated and glucuronide berberine in the brain after the oral administration of the Sucrosomial^®^ form. Glucuronide berberine plasma concentration was higher with Sucrosomial^®^ berberine compared to normal berberine. Finally, we observed similar increases of AMPK phosphorylation in the liver in response to the treatment with Sucrosomial^®^ berberine and berberine. *Conclusions*: The Sucrosomial^®^ formulation is an innovative and effective technology to improve berberine gastrointestinal (GI) absorption with proven in vitro and in vivo activity on insulin resistance.

## 1. Introduction

Berberine is a quaternary benzylisoquinoline alkaloid extracted from the root of the oriental *Berberis* plant *(B. aristata* and other species). It has a well-established hypocholesterolemic action by exerting, on average, a 10–20% lowering effect on low-density lipoprotein (LDL) cholesterol levels [1]. In addition, berberine administration reduces triglyceride (TG) and plasma glucose levels. For the first time, in 1988 the hypoglycemic effect of berberine was reported during the treatment of diarrhea in diabetic patients. Since then, berberine as an anti-diabetic agent has been used on large scale and is known as a folk medicine of China. A recent systematic review documented that berberine slightly lowers the glycated hemoglobin (HbA1c) level and the fasting plasma glucose (FPG) level, but remarkably reduces the 2 h post-load plasma glucose (2hPG) level when administered alone [2]. The mechanism of action of berberine that determines the hypocholesterolemic effect is related to the inhibition of the transcription of proprotein convertase subtilize/kexin type 9 (PCSK9) [3], and the stabilization of the LDL receptor mRNA [4]. Importantly, the inhibitory effect on PCSK9 expression by berberine seems to overcome the induction observed in response to monacolin K, thus supporting the combination with red yeast rice extracts [5]. Additional isoquinoline alkaloids, including californidine, show a berberine-like effect on PCSK9 expression [6,7].

Regarding the hypoglycemic effect of berberine, numerous studies have reported a metformin-like mechanism involving the activation of 5′-adenosine monophosphate (AMP)-activated protein kinase (AMPK) in response to an increase in the AMP/ATP ratio [8,9]. The inhibition of glucose oxidation in mitochondria may contribute to the AMP/ATP ratio increase and AMPK activation [10]. More recently, the anti-diabetic action of berberine has been ascribed to an increase in liver glucokinase (GK) and glycogen content in db/db mice [11]. This study confirmed a previous one conducted in alloxan-induced diabetic mice [12]. In more detail, berberine increased the dissociation of GK from GK regulatory protein (GKRP) and this was one of the potential mechanisms of hypoglycemic action of this alkaloid [11].

Berberine is well tolerated at doses up to 1.5 g daily although common gastrointestinal (GI) side effects can be observed. This safety profile is mainly due to its low solubility and low GI absorption, which also represents a limiting factor to its activity. The low bioavailability is due to the presence of positive charge on quaternary nitrogen and to the fact that berberine is a substrate of P-glycoprotein (P-gp), an efflux transporter that limits its absorption [13]. Thus, many efforts have been made to improve berberine’s oral bioavailability in order to reduce the administration of high dosages that result in GI adverse events [14,15].

To this aim, in the present study we provided new pharmacological evidence of efficacy and bioavailability of Sucrosomial^®^ berberine. This technology was originally developed to enhance the absorption of iron and to mitigate GI side effects [16,17]. However, its use for improving the GI absorption of natural alkaloids has never been documented.

## 2. Materials and Methods

### 2.1. Reagents

Eagle’s minimum essential medium (MEM) was purchased from Sigma (Milan, Italy); trypsin-EDTA, penicillin, streptomycin, sodium pyruvate, non-essential amino acid solution, fetal calf serum (FCS), plates and Petri dishes were purchased from EuroClone (Pero, Milan, Italy). 3,3’-Dioctadecyloxacarbocyanine perchlorate (DiO) was purchased from Sigma (Milan, Italy). Berberine and sucrosomial^®^ berberine were provided by PharmaNutra S.p.A. and were prepared by dissolving 10 mg of extracts in 1 mL of water for the in vitro studies; otherwise, a 15 mg/mL suspension was prepared in saline solution (0.9% NaCl).

### 2.2. Cell Cultures

Human hepatoma Huh7 cells were cultured in MEM supplemented with 10% FCS, L-glutamine, sodium-pyruvate and non-essential amino acids, penicillin/streptomycin at 37 °C in a humidified atmosphere of 5% CO_2_ and 95% air.

### 2.3. Cell Viability Assay

Cell viabilities were determined by sulphorhodamine B (SRB) assay, as previously described [18]. Cells were seeded in a 96-well tray (8 × 10^3^ cells/well) and after 48 h of incubation with increasing concentrations of nutraceutical compounds, the SRB assays were performed.

### 2.4. Retrotranscription and Quantitative PCR (RT-qPCR)

Total RNA was extracted with the iScriptTM RT-qPCR Sample Preparation Buffer cDNA synthesis preparation reagents (Bio-Rad S.r.l., Segrate, MI, Italy) according to the manufacturer’s instructions. Reverse transcription-polymerase first-strand cDNA synthesis were performed by using the Maxima First Strand cDNA Synthesis Kit (Thermo Scientific, Monza, MB, Italy) [19]. qPCR was then performed by using the PowerUp^TM^ SYBR^TM^ Green Master Mix (Thermo Scientific, Monza, MB, Italy) and specific primers for selected genes. The analyses were performed with the c1000 Touch qPCR System (Bio-Rad S.r.l., Segrate, MI, Italy), with the following cycling conditions: 95 °C, 2 min; 95 °C, 15 s and 60 °C, 1 min for 40 cycles. Data were expressed as Ct values and used for the relative quantification of targets with the ΔΔCt calculation. The ΔΔCt were made correct by multiplying the ratio value between the efficiency of the specific primer and housekeeping 18S.

### 2.5. Western Blot Analysis

Cells were washed twice with PBS and lysed with a solution of 50 mM Tris pH 7.5, 150 mM NaCl, 0.5% Nonidet-P40, containing protease and phosphatase inhibitor cocktails (Sigma S.r.l., Milan, Italy) for 30 min on ice. Then, 20 µg of proteins and a molecular mass marker (Thermo Scientific) were separated on 4–20% SDS-PAGE (Bio-Rad) under denaturing and reducing conditions. Proteins were then transferred to a nitrocellulose membrane by using the Trans-Blot^®^ Turbo™ Transfer System (Bio-Rad). Membranes were washed with Tris-buffered saline-Tween 20 (TBS-T), and nonspecific binding sites were blocked in TBS-T containing 5% non-fat dried milk for 60 min at room temperature. Blots were incubated overnight at 4 °C with a diluted solution (5% non-fat dried milk) of the following human primary antibodies: anti-GK (Biovision, Milpitas, CA, USA; rabbit polyclonal; dilution 1:1000), anti LDLR (Millipore, Darmstat, Germany; mouse monoclonal antibody, clone 2H7.1; dilution 1:1000), anti AMPK (GeneTex, Irvine, CA, USA; mouse monoclonal antibody, clone 34.2, dilution 1:1000 and phospho-AMPK (GeneTex, rabbit polyclonal antibodies, dilution 1:1000) and anti-β-actin (GeneTex, Irvine, CA, USA; rabbit polyclonal, dilution 1:1000) anti-α-tubulin (Sigma S.r.l., Segrate, MI, Italy; mouse monoclonal antibody, clone DM1A; dilution 1:5000). Membranes were washed with TBS-T and then exposed for 90 min at room temperature to a diluted solution (5% non-fat dried milk) of the secondary antibodies (peroxidase-conjugate goat anti-rabbit and anti-mouse, Jackson ImmunoResearch, Ely, Cambridgeshire, UK; dilution 1:5000). Immunoreactive bands were detected by exposing the membranes to Clarity^TM^ Western ECL chemiluminescent substrates (Bio-Rad, S.r.l., Segrate, MI, Italy) for 5 min, and images were acquired with a c100 Azure System (Aurogene, Rome, Italy). Densitometric readings were evaluated using the ImageLab^TM^ software (Bio-Rad S.r.l., Segrate, MI, Italy).

### 2.6. Animals and In Vivo Experimental Protocol

This investigation conformed to the European Commission Directive 2010/63/EU and was granted approval by the Italian Ministry of Health (587/2020-PR). C57BL/6 mice (*n* = 48; Charles River laboratories, Wilmington, MA, USA) were fed with standard diet for 7 days. On day 1, they were randomly subdivided into four groups of 12 mice each. The first group was fed with normal chow diet (standard diet, SD), the second with high fat diet (HFD) containing 45% kcal from fat, the third with HFD plus the administration by oral gavage of 50 mg/kg per day of berberine (Berberine-HCl, Shangay Freeman, Rijswijk, The Netherlands) and the fourth with HFD plus the administration by oral gavage of 50 mg/kg per day of Sucrosomial^®^ berberine (patent N° 102019000020290 owned by Alesco S.r.l., Pisa, Italy). Mice were fed with SD and HFD for a total of 16 weeks and the treatment with berberine and Sucrosomial^®^ berberine was performed during the last 8 weeks from Monday to Friday. At the end of the experimentation the animals were euthanized, and blood and organs collected. Plasma was separated by centrifugation at 5000× *g* at 4 °C and stored at −80 °C until analysis.

### 2.7. Quantification of Berberine and Its Metabolites in Plasma and Tissues by LC-MS/MS

The LC-MS/MS analysis of berberine and its metabolites in plasma and tissues was performed, improving upon the method that was used to assess pharmacokinetics of berberine after NADES administration [15].

Stock solutions were prepared by dissolving berberine and the benzanilide, used as the internal standard (ISTD), in methanol. The calibration curve was obtained by mixing 500 µL of 0.4 µg/mL ISTD with different volumes (500, 300, 100, 50, and 25 µL) of 100 ng/mL berberine standard solution in order to obtain different berberine/benzanilide ratios. A mixture of ISTD and berberine was added to blank plasma samples and to blank tissue and used for calibration curve preparation.

For plasma, 500 µL of IS solution in acidified methanol (5% formic acid) was added to 400 µL of whole blood to precipitate proteins. Solution was vortexed and centrifuged at 13,000 rpm for 15 min. The clear supernatant was concentrated under nitrogen flow at 25 °C. Then, 200 µL of methanol was used to dissolve the residue and the solution was used for LC-MS/MS. For analytical measurements, an Agilent series 1260 LC chromatograph coupled with a Varian 320 TQD MS spectrometer was used. The ion source was electrospray ionization (ESI) and analysis was conducted in positive ion mode. Analyses were performed on a Polaris 3 C18-A 150 × 3.0 mm (Agilent Technologies, Cernusco sul Naviglio, MI, Italy). The mobile phase was (A) water–formic acid (100:1.0 *v*/*v*) and (B) acetonitrile. A gradient program was used as follows: 0 → 1st min: A:B (70:30) isocratic, 1 → 7th min: A:B (70:30) → A:B (15:85) 7 → 15th min: A:B (15:85) → A:B (15:85), and re-equilibrating time A:B (70:30) for 5 min. The mobile phase flow rate was 0.3 µL/min. The injection volume was 5 µL. The ESI source was set in positive ionization mode. To quantify the metabolites, we used the reference work of Ma et al., 2013 using berberine as reference for quantitative purposes due to no commercially availability of other metabolite compounds.

Quantification was performed using multiple reaction monitoring (MRM) with m/z 336 > 291 transition for berberine, 338 > 321 for jatrorrhizine (reduced berberine), 322 > 307 jathorrizine (demethyl berberine), 324 > 308 demethylene-berberine, 514 > 338 jathorrizine-3-O-glucuronide, and *m*/*z* 198 > 105 transition for the ISTD. The MS parameters were capillary voltage 60 V, needle voltage 4200 V, shield voltage 600 V, collision energy 22 V, Q1 voltage 0.7 V and Q3 voltage 2.8 V, nebulizing gas pressure 50 psi and drying gas pressure 25 psi. Calibration curve using the ratio peak area berberine/peak area ISTD versus quantity berberine/quantity ISTD was y = 1.4415x + 0.0152, R2 = 0.9968. The limit of detection (LOD) was 0.2 ng/mL and the limit of quantification was 0.7 ng/mL. As described in Ma et al., a correction to the calibration curve of berberine was applied considering the molecular weight of the metabolites to obtain a semi-quantitative analysis of all the metabolites.

### 2.8. Statistical Analysis

Statistical analysis was performed using the Prism statistical analysis package Version 8.2.1 (GraphPad Software, San Diego, CA, USA). When possible, *p* values were determined by Student’s *t* test. Otherwise, differences between treatment groups were evaluated by one-way ANOVA. A probability value of *p* < 0.05 was considered statistically significant. The experiments were performed in triplicate.

## 3. Results

### 3.1. Determination of Sucrosomial^®^ Berberine Cytotoxicity in Huh7 Cells

The potential cytotoxic effect of Sucrosomial^®^ berberine was compared to berberine in Huh7 cells after 24 h incubation. The range of tested concentrations was between 2.5 and 40 µM. No significant cytotoxic effects were observed from both preparations (Figure 1).

### 3.2. Effect of Sucrosomial^®^ Berberine on AMPK and GK

Berberine is known to exert a significant glucose-lowering action and to improve insulin resistance both in clinical and experimental studies [19,20]. The main intracellular targets of berberine mediating this action are AMPK [8] and GK [11]. We, therefore, investigated the effect of Sucrosomial^®^ formulation on these two pathways by using Huh7 cells exposed to high glucose levels (25 mM). As shown in Figure 2, both berberine and Sucrosomial^®^ berberine induced the expression of GK and the phosphorylation state of AMPK. This effect was comparable between the two formulations, suggesting that berberine is also active in vitro in the Sucrosomial^®^ form. The activation of the AMPK pathway as well as the induction of GK preludes an antidiabetic action of Sucrosomial^®^ berberine at a similar potency than berberine.

### 3.3. In Vivo Pharmacokinetic Profile of Sucrosomial^®^ Berberine

The main aim of the present study was the development of a new formulation of berberine with higher GI absorption. Thus, we undertook a pharmacokinetic study before proceeding with the evaluation of the effect on insulin resistance in mice fed with HFD. We thus compared Sucrosomial^®^ berberine and berberine by the administration by oral gavage at the dose of 50 mg/Kg. After three weeks of treatment, we determined the plasma concentration of berberine in blood three hours after the last oral administration. Sucrosomial^®^ formulation determined a plasma concentration of berberine approximately 8-times higher than berberine (24 ± 17 ng/mL vs. 212 ± 87 ng/mL for berberine and Sucrosomial^®^ berberine, respectively). Based on these results, we reduced the dose of Sucrosomial^®^ berberine by 8 times and determined its effect on insulin resistance at the dose of 6.25 mg/kg compared to 50 mg/kg of control berberine.

Berberine was found to be metabolized by phase 1 reactions (reduction and de-methylation) followed by glucuronidation of demethyl derivatives (phase 2 reactions) (Figure 3), showing a pattern of metabolites in agreement with the literature [21]. The main metabolites observed were the dimethyl derivative jatrorrhizine, the demethylene berberine and the jathorrizine-3-O-glucuronide. Compounds were identified on the basis of their MS/MS spectra and in comparison with the literature [21].

The concentrations of berberine and its metabolites were measured in plasma and peripheral tissues. Berberine plasma concentrations were very similar after the oral administration of Sucrosomial^®^ berberine (6.25 mg/kg) and berberine (50 mg/kg) (Figure 4A). Importantly, we found considerable amounts of berberine in many tissues, including liver, kidney, visceral adipose tissue (VAT), heart and brain. These concentrations were normalized with total protein content of tissue extracts and cannot be compared to those observed in plasma. As expected, the highest amount of berberine was found in stools, thus supporting that the compound, due to its poor solubility, largely remains in cecal content, confirming its relatively poor bioavailability (Figure 4A).

On the contrary, significant differences of berberine metabolites were found comparing the two formulations (Figure 4B–D). The reduced berberine accumulated more significantly in the brain and VAT after oral administration of Sucrosomial^®^ form (Figure 4B). In the brain there was a tendency of accumulation of additional metabolites, such as demethylated and glucuronide berberine, with the latter reaching a much higher concentration in the plasma (Figure 4C,D).

### 3.4. In Vivo Efficacy of Sucrosomial^®^ Berberine on Insulin Resistance

As documented by the present study, Sucrosomial^®^ berberine is approximately 8 times more bioavailable than berberine, since the plasma concentrations were similar when administered at 6.25 mg/kg/day and 500 mg/kg/day, respectively. Thus, we next decided to investigate the effect on this new formulation on insulin resistance in vivo. C57BL/6 mice were fed HFD for 16 weeks and were treated by oral gavage with Sucrosomial^®^ and normal berberine during the last 8 weeks. As expected in response to HFD we observed a significant increase in body weight compared to SD (Figure 5). The administration of Sucrosomial^®^ and berberine did not affect the increase of body weight in these mice (Figure 5).

At the end of 16 weeks of HFD administration, mice developed insulin resistance, as documented by higher fasting glucose levels and during the OGTT test. The area under the curve (AUC) of glucose levels (mg/dL) vs. time (min) was equal to 48,548 ± 4756 mg/dL∙min and 73,103 ± 8645 mg/dL∙min for SD-treated and HFD-treated mice, respectively (Figure 6). Both Sucrosomial^®^ and non-formulated berberine significantly improved the insulin resistance as indicated by a reduction of AUC during the OGTT (58,830 ± 5597 and 51,518 ± 1984 mg/dL∙min for Sucrosomial^®^ and non-formulated berberine, respectively) (Figure 6). Thus, Sucrosomial^®^ berberine showed a similar activity than berberine but at 8-fold lower dose.

We than investigated the effect of Sucrosomial^®^ berberine on AMPK phosphorylation in the liver. Importantly, Sucrosomial^®^ and non-formulated berberine showed a similar increase of hepatic AMPK phosphorylation in C57BL/6 mice (Figure 7).

## 4. Discussion

In the present study we described a new formulation of berberine, named Sucrosomial^®^ berberine, that covers the alkaloid with a layer of phospholipids plus sucrose esters of fatty acids matrix [17]. This formulation has been extensively investigated for improving the absorption of Fe^3+^ and other minerals across the intestinal epithelium and has been shown to bypass the DMT-1 carrier, as it is physiologically involved in its GI absorption [22,23]. In addition, there is some evidence of the enhancer properties of sucrose esters for the accumulation of drugs in Caco-2 cells [24] and for intestinal permeability in animals [25]. However, its use in the administration of oral active ingredients has not been extensively studied. Here, we observed an 8-fold higher plasma concentration of Sucrosomial^®^ berberine compared to berberine hydrochloride when dissolved in physiological solution and administered by oral gavage at 50 mg/kg/day. This improved GI absorption was also associated with a significant in vivo efficacy in controlling glucose levels as well as the activation of hepatic AMPK phosphorylation.

Thus, 6.25 mg/kg/day of Sucrosomial^®^ berberine was as effective as 50 mg/kg/day of the hydrochloride one. However, important differences were found in the berberine metabolism. The results of the metabolite contents in plasma and tissues suggest that Sucrosomial^®^ formulation induced a higher phase 2 metabolization than control berberine. In fact, considerably higher plasma concentration of berberine glucuronide was observed in animals treated with Sucrosomial^®^ berberine. On the other hand, differences in the concentration and distribution of reduced berberine were evident with higher contents in the brain and VAT of Sucrosomial^®^ berberine-treated animals. Previous work demonstrated that iron-loaded (Sucrosome) vesicles could pass intact through the intestinal tissue. Thus, it is tempting to speculate that this “protective” action of Sucrosomial^®^ matrix may change the intestinal biotransformation of berberine including the enterohepatic circulation [26], thus determining a higher plasma concentration of the glucuronide form that can be transported across the blood–brain barrier, although this process is considerably slower than berberine [27]. It is also important to consider that different UDP-glucuronosyltransferases (UGT) isoforms are expressed in brain tissues to various levels and are known to participate to xenobiotic metabolism in the CNS [28].

Whether this altered metabolism could have an impact on the activity/toxicity of berberine still needs to be fully addressed. However, we observed a similar activity on improving insulin resistance induced by HFD compared to berberine hydrochloride. In addition, the dose of 6.25 mg/kg/day of Sucrosomial^®^ berberine was equally well tolerated compared to 50 mg/kg/day of berberine hydrochloride. Lack of toxicity is also proven by the fact that treated mice showed a similar body weight gain than the controls.

Our in vitro experiments, conducted on Huh7 hepatoma cells, also indicate that berberine incapsulated inside the Sucrosomial^®^ matrix is released and is bioavailable to act on its intracellular targets, i.e., AMPK. It is still not completely clear how the Sucrosomial^®^ is enzymatically digested inside the cells, although it has been demonstrated to be gastro-resistant and that Microfold cells of the Peyer’s patches (M cells) are involved in its intestinal absorption [23]. Thus, Sucrosomial^®^ formulation has been shown to ameliorate iron and magnesium GI absorption [23,29]. However, this approach seems to have a considerably higher impact on berberine GI absorption. Indeed, we observed 8-fold higher plasma concentration with Sucrosomial^®^ compared to berberine hydrochloride, while for iron and magnesium the GI absorption improved by approximately 2-fold [23,29]. These results highlight the negative role of the positively charged quaternary nitrogen atom in the berberine chemical structure on its bioavailability. The phospholipid bilayer membrane of Sucrosomial^®^ is capable of masking the positive charge and likely facilitating the GI absorption of berberine by a para-cellular and trans-cellular route [30,31].

Many other approaches have been pursued in order to improve the berberine bioavailability, including Huang-Gui Solid Dispersion [32], natural deep eutectic solvents [15], polysaccharide-based nanoparticles [33], and Phytosome^®^ technology [34]. The latter showed a 4–6-fold increase of berberine plasma concentration in healthy volunteers, which is very similar to our approach, and no major side effects were observed, thus anticipating a good safety profile of berberine. However, in our study, in order to avoid any unexpected side effects of berberine we decided to compare the daily dose of Sucrosomial^®^ berberine and berberine hydrochloride that determine the same plasma concentrations. This approach has permitted us to develop a new formulation of berberine that shows an effective and superimposable hypoglycemic action compared to berberine hydrochloride but at 8-fold lower concentration.

A second approach that has been recently developed to improve berberine activity consists of the use of liver-targeting nanotechnology that selectively delivers the alkaloid into the hepatocytes. This approach improved the efficacy of berberine on AMPK phosphorylation and ameliorated insulin-resistant status in HFD mice [35]. In our study, the determination of the concentration of berberine and its metabolites in the liver demonstrated that the Sucrosomial^®^ formulation did not significantly improve the uptake of the alkaloid in the hepatocyte. Berberine demethylated was the only metabolite that showed a significantly higher hepatic concentration after administration of the Sucrosomial^®^ formulation. However, this change did not improve the efficacy of this formulation on controlling insulin resistance.

## 5. Conclusions

For the first time, here we described a new formulation of berberine based on Sucrosomial^®^ technology with higher GI absorption that shows similar efficacy on controlling insulin resistance on mice fed with HFD as berberine hydrochloride, but at an 8-times lower dosage. Future studies in healthy volunteers will permit us to determine the effect of Sucrosomial^®^ berberine on glucose levels.

## Figures and Tables

**Figure 1 nutrients-14-03595-f001:**
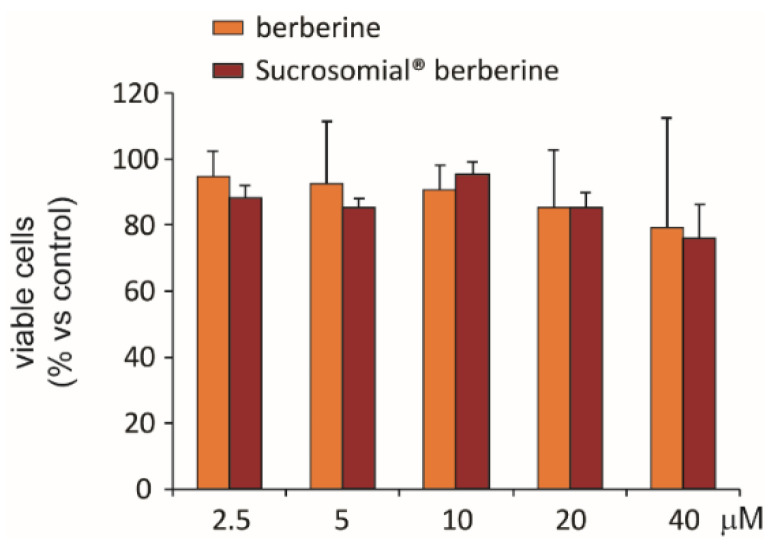
Cytotoxic effect of berberine and Sucrosomial^®^ berberine on Huh7 cell line. Huh7 cells were incubated for 24 h with indicated concentrations of berberine and Sucrosomial^®^ berberine. Cell viability was then determined by SRB assay.

**Figure 2 nutrients-14-03595-f002:**
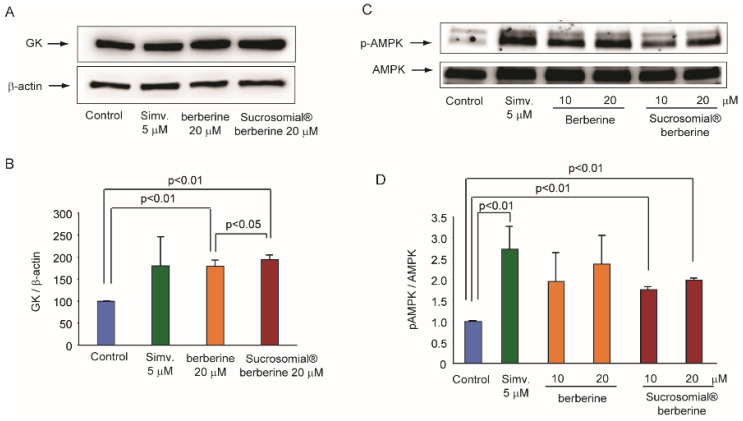
Effect of Sucrosomial^®^ berberine on GK expression and AMPK phosphorylation (**B**) in Huh7 cell line. Cells were incubated in DMEM containing 25 mM glucose with indicated concentrations of berberine for 24 h. After this period, total cellular proteins were extracted for the analysis. β-actin was used as loading control. (**A**) Western Blot of GK intracellular expression upon treatments; (**B**) Densitometric analysis of GK intracellular expression relative to β-actin; (**C**) Western Blot of pAMPK intracellular expression upon treatments; (**D**) Densitometric analysis of pAMPK intracellular expression relative to AMPK. DMEM: Dulbecco’s modified Eagle Medium; GK: glucokinase; AMPK: 5′ adenosine monophosphate-activated protein kinase; pAMPK: phospo-AMPK. DMEM, GK, AMPK.

**Figure 3 nutrients-14-03595-f003:**
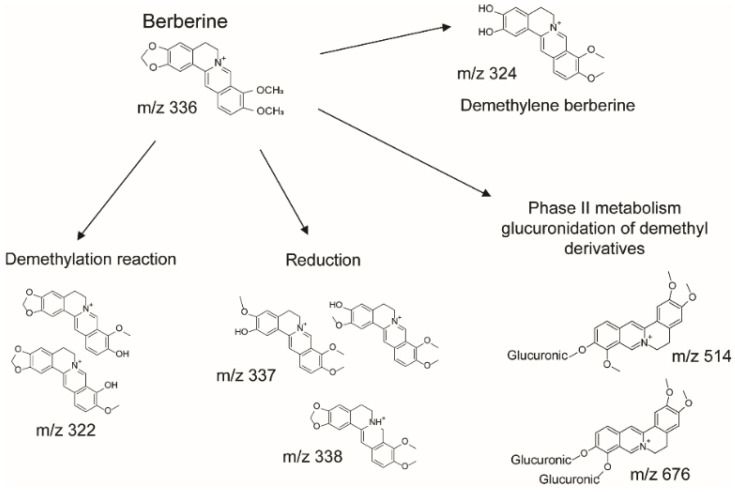
Metabolites detected after oral administration of Sucrosomial^®^ berberine.

**Figure 4 nutrients-14-03595-f004:**
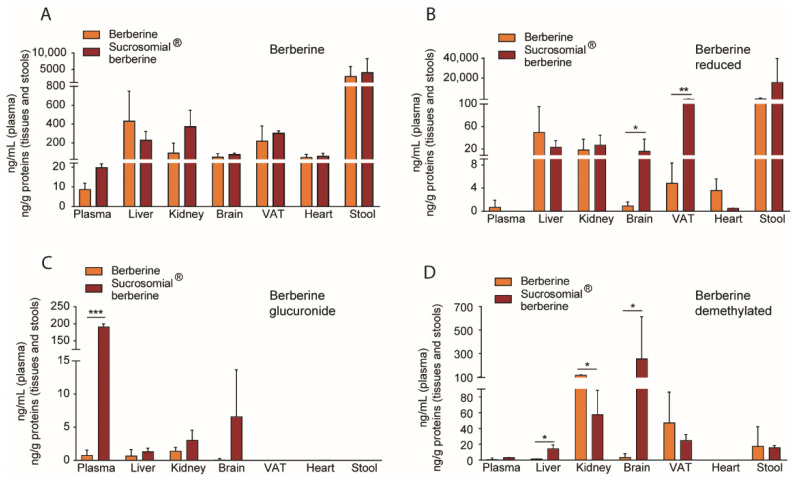
Distribution of berberine and its metabolites after oral administration of Sucrosomial^®^ berberine (6.25 mg/Kg) and berberine (50 mg/Kg). The Sucrosomial^®^ formulation triggers a differential berberine biodistribution among the analyzed tissues (**A**), as well as a differential biodistribution of the berberine metabolites such as the berberine reduced form (**B**), the berberine glucuronide (**C**), and the demethylated berberine product (**D**). VAT: visceral adipose tissue. * *p* < 0.05, ** *p* < 0.01, *** *p* < 0.001.

**Figure 5 nutrients-14-03595-f005:**
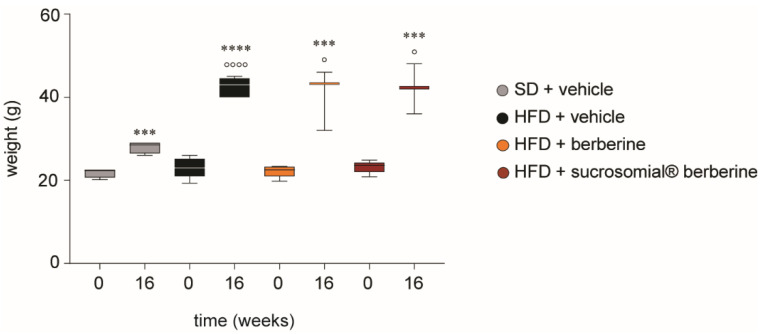
Effect of HFD and Sucrosomial^®^ berberine treatment on body weight of C57BL/6 mice during the 16 weeks of treatment. *** *p* < 0.001; **** *p* < 0.0001 vs. time 0. ° *p* < 0.05; °°°° *p* < 0.0001 vs. SD at 16 weeks.

**Figure 6 nutrients-14-03595-f006:**
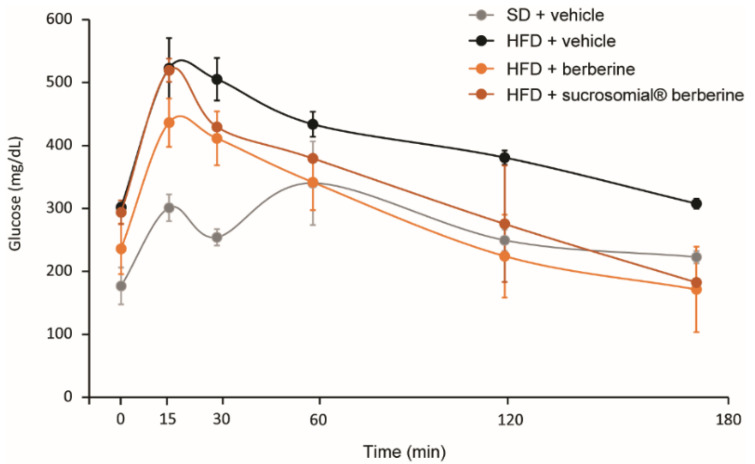
Effect of HFD and Sucrosomial^®^ berberine treatment on glucose levels during OGTT test in C57BL/6 mice after 16 weeks of diet and 8 weeks of treatment. HFD: high-fat diet; OGTT: oral glucose tolerance test.

**Figure 7 nutrients-14-03595-f007:**
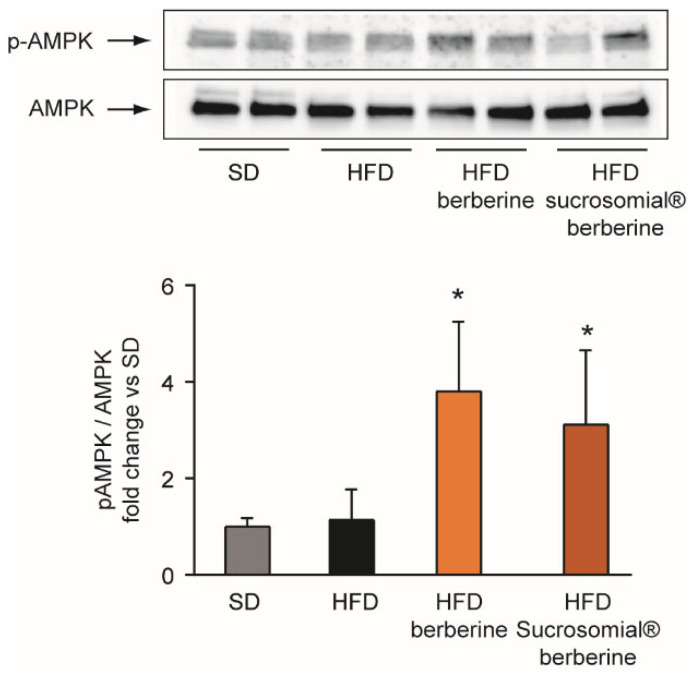
Effect of HFD and Sucrosomial^®^ berberine treatment on AMPK phosphorylation in the liver of C57BL/6 mice after 16 weeks of diet and 8 weeks of treatment. * *p* < 0.05 vs. HFD. The Western blot image is representative of the total samples analyzed (*n* = 12 per group), while the histograms show the results expressed as mean ± SE from all samples.

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
