# Peer review of "In Vitro and In Vivo Sucrosomial® Berberine Activity on Insulin Resistance"

_nutrients, 2022, doi:10.3390/nu14173595_

Round 1

Reviewer 1 Report

Dear Authors,

Your work is really interesting. I wonder if similar results would have been obtained in humans and I hope this will be the next step in your research.

I found no major issues in your work except for minor typing errors:

line 48: plasma glucose levels sounds better than glucose plasma levels

line 260: 6.25 mg/kg/day and 50 mg/kg/day instead of 6.25 mg/die and 500 mg/die

That's all my remarks.

Best Regards.

Author Response

Comment: Your work is really interesting. I wonder if similar results would have been obtained in humans and I hope this will be the next step in your research.

Answer: we thank the reviewer for the comment. As indicated in the discussion, there is one previous study that utilized a similar approach to improve berberine bioavailability, such as  Phytosome® technology 1. The authors showed a 4-6-fold increase of berberine plasma concentration in healthy volunteers, without any major side effects, anticipating a good safety profile of berberine. We are planning, in the future, to test our technology in healthy volunteers as well, but by comparing a 500 mg of regular berberine vs 60 mg (8 fold lower) Sucrosomial® berberine. We are predicted to see a very similar plasma concentration of berberine between the two formulations and likely similar glycemic control.

[1] Petrangolini, G, Corti, F, Ronchi, M, et al., Development of an Innovative Berberine Food-Grade Formulation with an Ameliorated Absorption: In Vitro Evidence Confirmed by Healthy Human Volunteers Pharmacokinetic Study, Evidence-based complementary and alternative medicine : eCAM, 2021;2021:7563889.

Comment: I found no major issues in your work except for minor typing errors:

line 48: plasma glucose levels sound better than glucose plasma levels

line 260: 6.25 mg/kg/day and 50 mg/kg/day instead of 6.25 mg/die and 500 mg/die

 Answer: We thank the reviewer, we have made these changes in the manuscript.

Reviewer 2 Report

The manuscript of “In vitro and in vivo Sucrosomial® Berberine Activity on Insulin 2 Resistance” shows the bioavailability advantage of Sucrosomial® berberine. A few comments:

1.       Line 155: The calibration curve was obtained mixing ISTD with different volumes (500, 300, 100, 50, and 25) of 100 ng/mL berberine standard solution. What is the unit of berberine volume?

2.      To quantify the metabolites, berberine was used as internal standard for quantitative purposes of other metabolite compounds. However, how to know the exact berberine amount when you used it as IS to quantify other metabolite, since the berberine has two sources, administered by oral gavage and added as IS during sample preparation. How do the authors calculate the amount of berberine derived from the oral gavage? Please clarify the confusion.

3.       This question is about the quantification.  Benzanilide was used as IS to quantify berberine, why can it be used as IS for other metabolites quantification?

4.       Line 196-197. Should be figure 1.

Author Response

Comment: Line 155: The calibration curve was obtained mixing ISTD with different volumes (500, 300, 100, 50, and 25) of 100 ng/mL berberine standard solution. What is the unit of berberine volume?

Answer: We thank reviewer for the comment, and we apologise if our description was not complete, we added the missing information in the text. The unit of berberine was microliters.

Comment: To quantify the metabolites, berberine was used as internal standard for quantitative purposes of other metabolite compounds. However, how to know the exact berberine amount when you used it as IS to quantify other metabolite, since the berberine has two sources, administered by oral gavage and added as IS during sample preparation. How do the authors calculate the amount of berberine derived from the oral gavage? Please clarify the confusion.

Answer: We thank reviewer for the comment. The sample preparation, as described in materials and methods, deal with protein precipitation by the treatment with acidified methanol. In order to minimise errors in the measurement an Internal Standard is added and this compound is added in exact amount. As stated in lines 153 the internal standard is Benzanilide. We modified the sentence in order to be sure that is clear for all the readers. There is no adding of berberine to samples, so the amount of berberine is correctly measured. The modified sentence is: “Stock solutions were prepared dissolving berberine and the benzanilide, used as Internal Standard (ISTD) in methanol. The calibration curve was obtained mixing 500 µL of 0.4 µg/mL ISTD with different volumes (500, 300, 100, 50, and 25 µL) of 100 ng/mL berberine standard solution in order to obtain different berberine/benzanilide ratios.”

Comment: This question is about the quantification.  Benzanilide was used as IS to quantify berberine, why can it be used as IS for other metabolites quantification?

Answer: We thank the reviewer for the comment. The ISTD is used as internal reference in the different samples, calibration is generated preparing solutions that contains ISTD and berberine and the calibration curve that is generated is used to calculate the amount of the berberine and the derivatives. All is described in lines 153-157. The calibration was used also for the measurement of the other metabolites as described in the lines 171-173 using an approach previously published by Ma JY, Feng R, Tan XS, Ma C, Shou JW, Fu J, Huang M, He CY, Chen SN, Zhao ZX, He WY, Wang Y, Jiang JD. Excretion of berberine and its metabolites in oral administration in rats. Journal of pharmaceutical sciences. 2013;102(11):4181-4192. Detailed description of the used protocol adopted for quantification is reported in the paper at lines 174-185 and reads “Quantification was performed using multiple reaction monitoring (MRM) with m/z 336>291 transition for berberine, 338>321 for jatrorrhizine (reduced berberine), 322>307 jathorrizine (demethyl berberine), 324>308 demethylene-berberine, 514>338 jathorrizine-3-O-glucuronide, and m/z 198 > 105 transition for the ISTD. The MS parameters were capillary voltage 60 V, needle voltage 4200 V, shield voltage 600 V, collision energy 22 V, Q1 voltage 0.7 V and Q3 voltage 2.8 V, nebulizing gas pressure 50 psi and drying gas pressure 25 psi. Calibration curve using the ratio peak area berberine/peak area ISTD versus quantity berberine/quantity ISTD was y = 1.4415x + 0.0152, R2 = 0.9968. The limit of detection (LOD) was 0.2 ng/mL and the limit of quantification was 0.7 ng/mL. As described in reference of Ma et al., a correction to the calibration curve of berberine were applied considering the molecular weight of the metabolites to obtain a semiquatitative analysis of all the metabolites.

Comment: Line 196-197. Should be figure 1.

Answer: We thank reviewer. We have changed the figure number.

Round 2

Reviewer 2 Report

In the response, "The ISTD is used as internal reference in the different samples, calibration is generated preparing solutions that contains ISTD and berberine and the calibration curve that is generated is used to calculate the amount of the berberine and the derivatives.". However, in line 153-154, the authors never mentioned there were derivatives in the calibration samples.  

Author Response

Comment: "The ISTD is used as internal reference in the different samples, calibration is generated preparing solutions that contains ISTD and berberine and the calibration curve that is generated is used to calculate the amount of the berberine and the derivatives.". However, in line 153-154, the authors never mentioned there were derivatives in the calibration samples.  

Response: We thank the reviewer for the comment, and we apologise if the answer has not been clear. As stated by the reviewer we never reported that derivatives were added in the calibration curve. Indeed, calibration is generated by preparing solutions containing ISTD and berberine and the calibration curve that is generated is used to calculate the amount of the berberine and derivatives”. This aspect is also reported in the lines 171-173. “To quantify the metabolites, we used the reference work of Ma et al., 2013 using berberine as reference for quantitative purposes due to non-commercially availability of other metabolite compounds.”

Round 3

Reviewer 2 Report

The authors have provide sound response.